

**Enabling a process-oriented hydro-biogeochemical model to simulate soil**
**erosion and nutrient losses**
Siqi Li [a, b], Bo Zhu [c], Xunhua Zheng [a, d], Pengcheng Hu [c], Shenghui Han [a], Jihui Fan [c], Tao Wang
[c], Rui Wang [a], Kai Wang [a], Zhisheng Yao [a], Chunyan Liu [a], Wei Zhang [a, *], Yong Li [a, *]
[a] State Key Laboratory of Atmospheric Boundary Layer Physics and Atmospheric Chemistry,
Institute of Atmospheric Physics, Chinese Academy of Sciences, Beijing 100029, China
[b] State Environmental Protection Key Laboratory of Formation and Prevention of Urban Air
Pollution Complex, Shanghai Academy of Environment Sciences, Shanghai 200233, P. R.
China
[c] Institute of Mountain Hazards and Environment, Chinese Academy of Sciences, Chengdu
610041, China
[d] College of Earth and Planetary Science, University of Chinese Academy of Sciences, Beijing
100049, China
[*] Corresponding author
Tel.: +86 10 13681042146

16   +86 10 13107488562

E-mail address: zhangwei87@mail.iap.ac.cn (W. Zhang)
yli@mail.iap.ac.cn (Y. Li)



## Abstract

**Abstract**
Water-induced erosion and subsequent particulate carbon (C), nitrogen (N) and phosphorus
(P) nutrient losses were the vital parts of biogeochemical cycling. Identifying their intensity and
distribution characteristics is of great significance for the control of soil and water loss and N/P
nonpoint source pollution. This study incorporated the modules of physical soil erosion and the
particulate C, N and P losses into the process-oriented hydro-biogeochemical model
(CNMM-DNDC) to enable it to predict soil and water loss. The results indicated that the
upgraded CNMM-DNDC i) performed well in simulating the observed temporal dynamics and
magnitudes of surface runoff, sediment and particulate N/P losses in the lysimetric plot of the
Jieliu catchment in Sichuan Province; ii) successfully predicted the observed monthly dynamics
and magnitudes of stream flow, sediment yield and particulate N losses at the catchment outlet,
with significant zero-intercept univariate linear regressions and credible Nash–Sutcliffe indices
larger than 0.74. The particulate N accounted for 16.2%−26.6% of the TN components during
the period with larger precipitations. The intensities of soil erosion and particulate nutrient
losses in the Jieliu catchment was closely related to land use type in the order of sloping
cultivated cropland > residential area > forest land. The scenario analysis demonstrated that
high greenhouse gas (GHG) emissions scenarios provided a greater risk of soil erosion than did
low GHG emissions scenarios and land use change could help to mitigate soil and water loss
accelerated by climate change in the future. The upgraded model was demonstrated to have the
capability of predicting ecosystem productivity, hydrologic nitrogen loads, emissions of GHGs
and pollutant gases, soil erosion and particulate nutrient losses, which may become a decision





support tool for soil erosion and nonpoint source pollution control coordinated with increasing
production and reducing GHGs and pollutant gases emissions in a catchment.

**Keywords**

CNMM-DNDC, ROSE, soil erosion, particulate carbon/nitrogen/phosphorus loss

## 1. Introduction

Water-induced erosion and subsequent particulate carbon (C), nitrogen (N) and
phosphorus (P) nutrient losses are among the primary threats leading to the decline in soil
fertility and the increases in land degradation, channel sedimentation and eutrophication of
downstream rivers and lakes (Berhe et al., 2018; Ekholm and Lehtoranta, 2012; Garcia-Ruiz et
al., 2015). This global environmental issue has continued to deteriorate (Ma et al., 2021; Yang
et al., 2003). A previous study found that the vulnerability of water-induced erosion increased
over 51% of the global surface from 1982 to 2015 (Liu et al., 2019). Climate change and
anthropogenic activities (such as land use change) are the two principal driving forces that have
complicated and altered the hydrological cycle and water-induced erosion during recent decades
(Piao et al., 2007; Zeng et al., 2015).
Quantitative assessments of the water-induced soil erosion intensity and identification of its
temporal and spatial distribution characteristics are of great importance for preventing soil and
water loss and have attracted the attention of researchers (e.g., Jetten et al., 2003; Jiang et al.,
2017; Panagos et al., 2015). Lysimetric plot experiments have been developed as a direct field
measurement method for the accurate quantification of surface runoff and water-induced
erosion (e.g., Kosmas et al., 1997; Sumner et al., 1996; Zhu et al., 2009). However, the in situ
field measurements of water-reduced soil erosion with high cost of labor and money can only



cover a small piece of the sampling units. It is unrealistic to expect direct field measurements
to quantify water-induced erosion everywhere under various conditions.

Simulations of mathematical models are likely to compensate for the deficiency of direct

field measurements. The Universal Soil Loss Equation (USLE, Wischmeier and Smith, 1978)
and its revised version (RUSLE) (Renard et al., 1997) have been developed into widely used
empirical mathematical models to directly calculate soil erosion based on rainfall, soil property,
topography, cover and management data. The USLE or RUSLE quantify only the various
influencing factors that impact the soil loss associated with soil erosion, which is not directly
related to the process of surface runoff and does not involve the specific process of sediment
transport yet (Donovan, 2022; Meinen and Robinson, 2021). Fortunately, the physical
process-based ROSE model named after the name of developer (Rose et al., 1983)
conceptualizes the soil erosion process by conceiving three continuous and simultaneous
physical processes, including rainfall detachment, sediment entrainment and sediment
deposition, thus providing good performance in estimating sediment yield at the plot scale.
However, the ROSE model focuses only on the physical processes of water-induced erosion
without engaging the C and N cycles of the ecosystem. The Soil and Water Assessment Tool
(SWAT) (Arnold et al., 1998), a semi distributed hydrological model, incorporates the RUSLE
to predict soil erosion at the level of hydrological response units, in which the routing of
sediment transportation is not considered and the modeling of the biogeochemical element
cycle is relatively simple and empirical (Ferrant et al., 2011; Pohlert et al., 2007). However, the
transport of particulate C, N, and P nutrients accompanied by water-induced erosion crucially





depends on the C and N cycles of ecosystems in a catchment. Therefore, knowledge of the
coupling between the process-oriented hydro-biogeochemical model combined with the
complicated C and N cycles and the soil erosion model based on physical processes (such as
ROSE) is essential to accurately predict soil erosion and subsequent particulate C, N, and P
nutrient transport.

A recently developed hydro-biogeochemical model (CNMM-DNDC) by Zhang et al.

(2018) might become a realistic tool that can be used to address the abovementioned problem.
The CNMM-DNDC model introduces the complicated C and N biogeochemical modules
(including the modules of decomposition, nitrification, denitrification and fermentation) of a
widely used biogeochemical model (DeNitrification-DeComposition model, DNDC, Li et al.,
1992) into the distributed hydrological framework of the Catchment Nutrients Management
Model (CNMM, Li et al., 2017). The CNMM-DNDC model has been used to conduct a
comprehensive simulation of the complicated hydrological and biogeochemical processes
(such as ecosystem productivity, hydrologic N loads, gaseous N losses and greenhouse gas
emissions) of a subtropical catchment with various landscapes (Zhang et al., 2018), a model
evaluation of nitrous oxide and nitric oxide emissions from a subtropical tea plantation (Zhang
et al., 2020b), a model evaluation and regional simulation of nitrate leaching in the black soil
region of Northeast China (Zhang et al., 2021a) and a comprehensive model modification and
evaluation of $NH_3$ volatilization from fertilized croplands (Li et al., 2022b). However, the
CNMM-DNDC model still lacks the capacity to simulate the processes of soil erosion and
subsequent particulate C, N, and P nutrient transportation.





Therefore, we hypothesize that the accurate simulation of soil erosion and subsequent
particulate C, N, and P nutrient losses can be realized by incorporating the soil erosion physical
model and the element enrichment module into the process-oriented hydro-biogeochemical
model with complicated C and N cycles. Based upon the above hypothesis, the objectives of
this study were to i) introduce the ROSE model (a physical soil erosion model) and the
enrichment module of the particulate nutrients into the hydrological process of the
CNMM-DNDC model; ii) evaluate the performance of the CNMM-DNDC model in simulating
the temporal and spatial distributions of soil erosion and subsequent particulate C, N, and P
transportation at the plot and catchment scales; and iii) investigate the impact of climate change
and human activities (such as land use change) on the losses of soil and particulate nutrients.
**2. Materials and methods**
**2.1 Catchment description**

The Jieliu catchment (31°16′N, 105°28′E, 400−600 m a.s.l.), located in Sichuan Province

of Southwest China (Zhu et al., 2009), was used for the model calibration and validation. This
catchment is situated in the upper reaches of the Yangtze River and has a typical subtropical
monsoon climate. During the period from 2005 to 2018, the annual mean temperature was
16.7 ℃, and the average annual precipitation was 720 mm, 75% of which occurred during the
period between June and September (http://yga.cern.ac.cn). The soil in the catchment is
dominated by Calcaric purple soil, classified as a Pup-Orthic Entisol in the Chinese Soil
Taxonomy or as an Entisol classified in the U.S. Soil Taxonomy (Zhu et al., 2009). The total
area of the Jieliu catchment is approximately 35 ha, and it is dominated by sloping croplands
(54%) and forest lands (31%). The primary crops cultivated in the sloping croplands are maize



(Zea mays L.), winter wheat (Triticum aestivum L.), rape (Brassica napus L.) and rice (Oryza
sativa L.). The N, P and potassium (K) fertilizers are applied at rates of 130–330 kg N ha$^{-1}$ yr$^{-1}$
(ammonium bicarbonate or urea), 72–162 kg P ha$^{-1}$ yr$^{-1}$ (calcium superphosphate) and 45–68
kg K ha$^{-1}$ yr$^{-1}$ (potassium chloride), respectively (Zhang et al., 2018).
**2.2 Model modifications**

The CNMM-DNDC model can simulate the lateral movements of water-soluble nutrients

(e.g., ammonium, nitrate, phosphate and dissolved organic matter) by surface and subsurface
runoff, whereas it lacks the capabilities of simulating soil erosion and sediment transport
caused by surface runoff and the subsequent transportation of particulate C, N, and P. To
address such a deficiency, this study incorporated the modules of soil erosion and element
enrichment into the lateral hydrological framework of the CNMM-DNDC model. Therefore,
the upgraded CNMM-DNDC model was equipped with the ability to estimate the movements
of soil particles and particulate nutrients transported with surface runoff in the lateral
dimension. The soil erosion module adopted the simplified ROSE model (Rose et al., 1983;
Stewart, 1985), which is a process-oriented soil erosion model. The ROSE model is based on
the dynamic equilibrium of three simultaneous processes, including rainfall detachment, runoff
detachment, and sediment deposition. In an individual erosion event, the process of runoff
detachment dominates, and the latter two processes of rainfall detachment and sediment
deposition can be generally neglected. Therefore, in the simplified ROSE module, as shown by
Eq. (1), the sediment yield ($Y_s$, kg dry soil ha$^{-1}$) resulting from soil erosion was driven by
surface runoff ($R_s$, m) and concomitantly regulated by the land's slope ($S_l$, dimensionless) and





the coverage fraction of vegetation ($C_v$, fraction).

$$Y_s = 27 \times 10^6 (1 - C_v) \eta S_l R_s \tag{1}$$

Where $R_s$ is calculated from the existing hydrological module of the CNMM-DNDC
model, in which $R_s$ occurs in the following two cases. First, $R_s$ is caused by the mechanism of
excess infiltration, in which the water input (i.e., precipitation and irrigation) is greater than the
maximum infiltration capacity of the soil. Second, $R_s$ is derived from the mechanism of excess
storage, in which precipitation or irrigation still occurs when the soil surface water content
exceeds the corresponding saturated water content. The direction of the surface runoff conflux
is estimated by the distributed weights of four neighboring grids (i.e., in the upper, lower, left
and right directions), which are calculated based on the elevation of these grids. $S_l$ is the sine
value of the slope radian value. In this study, the value of $C_v$ for the crop is approximately
equivalent to the growing index, which is estimated by the ratio of the accumulated
temperature from sowing to the present time to the accumulated thermal degree for maturity in
the plant growth module. Particularly, using the observed sediment yield in the catchment
outlet, the value of $C_v$ for the natural vegetation (e.g., forest and grass) was addressed and
calibrated as half of the ratio of the real leaf area index (LAI) and the maximum LAI, which is
one of the model inputs. The value of $C_v$ of the artificial lands (e.g., the urban or rural
residential areas) was calibrated and set to 0.1, which represented the effects of concrete roads
and residential buildings on the reduction of the soil area exposed to erosion. Usually, the
values of $C_v$ need to be calibrated by the soil exposure ratio and sediment yield observations





for a given study area. $\eta$ (dimensionless) is referred to as the efficiency of sediment entrained
by surface runoff, which depends on soil texture and $C_v$, as shown in Eq. (2). In Eq. (2), $a_1$ is
referred to as the rate of sediment carried by surface runoff on bare land, which differs for
various soil types and generally needs to be calibrated by the observed data of sediment loss for
a given study area. Loch and Donnollan (1983) reported that $a_1$ varies from 1.0% to 8.7% in
Middle Ridge clay loam and Irving clay soils. The soil mass balance was not considered in the
upgraded model.

$$\eta = a_1 e^{-0.15 C_v} \tag{2}$$

It is known that the C, N, and P elements of the eroded sediments are usually richer than

those of the in situ soils from which the eroded sediments originate (Massey and Jackson, 1952;
Schiettecatte et al., 2008; Wan and El-Swaify, 1998). The above phenomenon is usually referred
to as sediment enrichment, which can be quantified by an empirically based enrichment ratio
($E$). $E$ is usually defined as the ratio of the concentration of C, N, and P elements in the eroded
sediment to that in the source soil (Sharpley, 1980; Teixeira and Misra, 2005). Generally, as
more eroded sediment is produced, the richness of the C, N, and P elements decreases. The
enrichment ratio of the C and N nutrients ($E_{CN}$) is estimated by Eq. (3), which was adapted
from McElroy et al. (1976) and Williams and Hann (1978). The pre-exponential factor ($k_1$) of
Eq. (3) was calibrated to 1.2 using the particulate N data observed at the lysimetric plot in this
study. The enrichment ratio of P nutrients ($E_P$) is calculated by Eq. (4) cited from Sharpley

(1980).





$$E_{CN} = k_1 (Y_s \times 10^{-4})^{-0.2468} \tag{3}$$

$$E_P = e^{(2.46 - 0.2 \log Y_s)} \tag{4}$$

The yields of particulate C ($P_C$, kg C ha$^{-1}$), N ($P_N$, kg N ha$^{-1}$), and P ($P_P$, kg P ha$^{-1}$)
nutrients caused by soil erosion were calculated based on $E$, $Y_s$ and the content of the
corresponding organic C ($C_C$, g C ha$^{-1}$), N ($C_N$, g N ha$^{-1}$), and P ($C_P$, g P ha$^{-1}$) pools in topsoil
using Eqs. (5−7), respectively. BD (g m$^{-3}$) and $D_s$ (m) refer to the soil bulk density and the
depth of topsoil, respectively. Eight of the soil organic C and N subpools participated in the
process of soil erosion, including the pools of very reliable, reliable, and resistant
decomposable litters, reliable and resistant active microbes, reliable and resistant humads and
passive humus, whereas five of the soil organic P subpools were involved in the process of soil
erosion, including the pools of active and passive organic P, active and dead microorganism P,
and inert stable P. The particulate C, N, and P losses calculated by the element enrichment
module were also deducted from the corresponding subpools of the topsoil. Subsequently, the
eroded soil and the particulate C, N, and P nutrients are transported with surface runoff and
eventually drain into streams.

$$P_C = \sum_{i=1}^{8} \frac{10^{-7} E_{CN} C_{C_i} Y_s}{BD \cdot D_s} \tag{5}$$

$$P_N = \sum_{j=1}^{8} \frac{10^{-7} E_{CN} C_{N_j} Y_s}{BD \cdot D_s} \tag{6}$$



$$P_\mathrm{P} = \sum_{k=1}^{5} \frac{10^{-7} E_\mathrm{P} C_{\mathrm{P}_k} Y_\mathrm{s}}{\mathrm{BD} \cdot D_\mathrm{s}} \tag{7}$$

**2.3 Preparation for model simulation**


The input data for driving the model operation consisted of the meteorological data at the
3-hour scale (including average air temperature, solar radiation, long wave radiation, wind
speed, humidity and total precipitation), the spatialized soil properties (including soil texture,
soil organic carbon, bulk density and pH), the gridded digital elevation model (DEM, Fig. 1)
with a resolution of 5 m ×5 m, the spatial distribution of land use (Fig. 1) and cropping systems,
and the field management practices. Taking the efficiency of the model calculation and the
accuracy of the biogeochemical process description into consideration, the upgraded
CNMM-DNDC model conducted a simulation with a grid of 15 m × 15 m from 2004 to 2017,
with an initial spin-up period of ten years. The DEM, soil properties, land use, cropping
systems, field management practices and meteorological data from 2004 to 2014 were
primarily adapted from Zhang et al. (2018). The remaining meteorological data were adapted
from the hourly observations provided at the National Science & Technology Infrastructure
(http://rs.cern.ac.cn). The input data of soil properties, DEM, land use, cropping systems, and
field management practices were resampled to the ASCII grids with a resolution of 15 m ×15
m using the ArcGIS 10.0 software package (ESRI, Redland, CA, USA). The observation data
measured at the lysimetric plot and the catchment outlet (Fig. 1) contributed to model
calibration and validation. The surface runoff, subsequent sediment yield, and particulate and
total N losses from 2004 to 2006 with three replicates and the surface runoff, subsequent
sediment yield, and total P loss from 2017 to 2018 with three replicates measured at the





lysimetric plots were adapted from Deng et al. (2011) and Hu (2020), respectively. The
monthly stream flow, sediment yield, and particulate and total N losses from 2007 to 2008
measured at the catchment outlet were directly cited from Deng et al. (2011). Total N referred to
the total amount of $NH_4^+$, $NO_3^-$, dissolved organic N and particulate N. Total P referred to the
total amount of dissolved organic and inorganic P and particulate P. Among them, the observed
data from the lysimetric plot in 2004 (with seven observation times) and 2016 (with four
observation times and a heavy precipitation event) and the observed data from the catchment
outlet in 2007 were used for model calibration, and the remaining observed data were used for
model validation.
**2.4 Climate and land use scenario settings**
Scenario analysis was adopted to assess the impact of climate change and land use change
on water-induced erosion and its accompanying nutrient losses. The annual accumulated yields
of sediment and particulate C, N, and P nutrient losses at the outlet in 2008 (the year for model
validation) simulated by the upgraded model were used as the baseline values of the scenario
analysis. Two groups of climate change and land use change scenarios were designed:
single-factor change and multifactor change scenarios. The single-factor change scenarios
altered only one factor while keeping the others constant. The single-factor change scenarios of
climate change consisted of two parts. One part for air temperature change was altered within
the range of −4 ℃ to +4 ℃ with an interval of 0.2 ℃ (abbreviated as $T_{air}+$ the increase value
of air temperature or $T_{air}-$ the decrease value of air temperature). The other part for
precipitation change was altered by the range from −30% to +30% with an interval of 2%



(abbreviated as $P+$ the increase percentage of precipitation or $P-$ the decrease percentage of
precipitation). A single-factor change scenario of land use was designed as the sloping upland
changed into forest land with the lower soil erosion rate (i.e., FL scenario). The existing land
use conversion to another type, such as the change from cropland to forest land or some other
land use, is a kind of compromise and required a sensitivity analysis to the model simulation
rather than representing the conditions of the real natural system (Dey and Mishra, 2017). The
multifactor change group was designed to simultaneously consider climate and land use change.
The IPCC's Summary for Policy-makers (IPCC, 2021) points out that the average annual
global land precipitation is projected to increase by 10.5% and 30.2% at the 1.5 ℃ and 4 ℃
warming levels, respectively. According to the correspondence between climate warming and
increasing precipitation in the IPCC's AR6, the multifactor change scenarios designed two
multiple climate change scenarios: the low and high greenhouse gas (GHG) emissions
scenarios. The low GHG emissions scenario represents air temperature and precipitation
increasing by 1.5 ℃ and 10%, respectively, while the high one represents air temperature and
precipitation increasing by 4 ℃ and 30%, respectively. For the sake of argument, we also
divided air temperature and precipitation single-factor scenarios into four sets. The scenarios
with air temperature increases greater than 2 ℃ were defined as the higher warming scenarios,
while the lower ones were defined as the scenarios with air temperature changes from 0 ℃ to
2 ℃. The scenarios with air temperature reductions greater than 2 ℃ were defined as the
higher cooling scenarios, while the lower cooling scenarios were defined as the scenarios with
air temperature changes from −2 ℃ to 0 ℃. Similarly, the scenarios with precipitation





increases greater than 20% were defined as the higher rain-enhanced scenarios, while the lower
ones were defined as the scenarios with precipitation changes from 0% to 20%. The scenarios
with precipitation decreasing more than 20% were defined as the higher rain-reduced scenarios,
while the lower rain-reduced scenarios were defined as the scenarios with precipitation change
from −20% to 0%. Furthermore, we also explored the effect of the low and high GHG
emissions scenarios in a combination of land use change scenarios (i.e., FL scenario) on
sediment yield and particulate nutrient yields. The tillage scenario analysis was involved in the
scenario analysis of alternative management practices, which were conducted as the no tillage
operations of the short term in 2008 and the long term from 2004 to 2008. The relative change
deviations of the simulated annual accumulated sediment and particulate nutrient losses of the
designed scenarios from the baseline were provided as the quantitative evaluation index for
scenario analysis (Abdalla et al., 2020; Dubache et al., 2019).
**2.5 Evaluation of model performance and statistical analysis**

The performance of the upgraded CNMM-DNDC model in simulating sediment and

particulate nutrient losses was evaluated using the normalized root mean square error (nRMSE),
the Nash–Sutcliffe index (NSI) and the slope, determination coefficient ($R^2$) and significance
level ($p$) of the zero-intercept univariate linear regression (ZIR) between the simulation and
observation. The nRMSE and NSI values are calculated by Eq. 8 and Eq. 9, respectively. $O_i$
and $S_i$ are the observed and simulated values, respectively. $\overline{O}$ is the mean value of the
observed data, and $n$ is the number of paired samples. If the value of nRMSE is closer to 1, the
values simulated by the model are more coincident with the observed values (Cui et al., 2014;





Smith et al., 1997). The value of the NSI provides the discrepancy between the simulated
values and the mean of the observed values, with a positive value indicating an acceptable
simulation (Li et al., 2022a). The closer to 1 the slope and $R^2$ of the ZIR are, the better the
simulated values match the observed values. The Origin 8.0 (OriginLab Ltd., Guangzhou,
China) and ArcGIS 10.0 software packages were used for graph drawing.

$$nRMSE = \frac{100}{\overline{O}} \sqrt{\frac{\sum_{i=1}^{n}(S_i - O_i)^2}{n}} \tag{8}$$

$$NSI = 1 - \frac{\sum_{i=1}^{n}(S_i - O_i)^2}{\sum_{i=1}^{n}(O_i - \overline{O})^2} \tag{9}$$

In addition, linear correlations were carried out to study the relationships between the

variables relevant to soil erosion and that related to the biogeochemistry process. The R project
was applied for the graph drawing of correlation matrix.
**3. Results**
**3.1 Model performance in simulating soil erosion in the lysimetric plot**

Among the only eight soil erosion observations conducted at the lysimetric plot from 2015

to 2017, four observations in 2016 were provided for model calibration. Given the limited size
of the samples, the performance of the upgraded CNMM-DNDC model was revealed using
only the graph of the predictions and observations (Fig. 2a−c), without a quantitative
evaluation with the above statistical criteria. The temporal dynamic patterns of the simulated
surface runoff, sediment and concomitant particulate P yields were in accordance with the
observed values when either model calibration or validation was performed (Fig. 2 a−c).
Nevertheless, on July 23, which was a heavy precipitation event (213 mm precipitation during



the seven days prior to the observation day) in 2016, the upgraded model overestimated the
observed sediment yield by approximately 6 times (3.6 versus 0.6 t ha$^{-1}$, Fig. 2b). However, the
simulated surface runoff and total P loss were only approximately 60% and 20% larger than the
observed values, respectively. Unfortunately, the simulated peaks of surface runoff and
sediment yield at the end of June 2015 lacked the support of the observations. Previous results
by the authors also demonstrated that the upgraded CNMM-DNDC performed well in
simulating surface runoff, sediment yield, and particulate N loss in 2004 (i.e., the verification
period) in the lysimetric plot, with significant ZIRs and credible nRMSE values of 15.2%,
32.0%, and 88.0%, respectively (Table S1 and Fig. 2d−g). Moreover, we conducted an
evaluation of the simulated and observed $NH_4^+$ and $NO_3^-$ losses accompanied by surface runoff
in the lysimetric plot (Fig. S1). The upgraded model generally captured the temporal variation
and magnitude of the observed $NH_4^+$ and $NO_3^-$ loss, although discrepancies existed in the
magnitude of the peak loss (i.e., the model underestimated $NH_4^+$ loss caused by approximately
100 mm precipitation on September 4, 2006; Fig. S1).
**3.2 Model performance in simulating soil erosion at the catchment outlet**

The monthly observed and simulated stream flow, sediment yield, particulate and total N

losses at the outlet of the Jieliu catchment from 2007 to 2008 are illustrated in Fig. 3. The
observed stream flow and sediment yield began to increase dramatically with the concentrated
precipitation in summer and early autumn but rarely occurred in winter and spring (Fig. 3a−c).
The upgraded CNMM-DNDC model successfully predicted the above temporal pattern of the
stream flow and sediment yield at the catchment outlet with credible NSI values of 0.89 and





0.89 and significant ZIRs with $R^2$ values of 0.94 and 0.92 and slope values of 0.85 and 0.88 for
model validation, respectively (Table 1). Moreover, model validation of sediment yield resulted
in a larger nRMSE (38.23%) than that of stream flow simulation (34.57%).

The observed particulate and total N losses revealed a similar temporal pattern to that of

sediment yield (Fig. 3d−e) ranged from 0 to 56.3 kg mon$^{-1}$ and 0.9 to 283.1 kg N mon$^{-1}$ with a
mean value of 10.5 and 55.9 kg N mon$^{-1}$, respectively. The corresponding simulated
particulate and total N losses resulted in ranges of 0.5 to 50.4 kg N mon$^{-1}$ and 18.8 to 196.0 kg
N mon$^{-1}$ with averages of 12.0 and 65.1 kg N mon$^{-1}$, respectively. The upgraded model
provided an overestimation of the particulate N loss in August 2007 and September 2008 by
11.3 and 14.8 kg N mon$^{-1}$, respectively. The particulate N losses in February 2007, March 2007,
July 2007 and July 2008 and total N loss in summer were underestimated. However, in terms
of the validation, statistical comparisons between the simulated particulate and total N losses
yielded significant ZIRs with $R^2$ values of 0.85 and 0.86 and slope values of 0.80 and 0.96,
nRMSE values of 57.75% and 42.55%, and NSI values of 0.74 and 0.86, respectively ($n = 12$;
Table 1). Meanwhile, the upgraded CNMM-DNDC model successfully predicted the temporal
variation and magnitudes of $NO_3^-$ loss at the catchment outlet, although the model slightly
underestimated the peak loss in July and August of 2007 and in September of 2008 (Fig. S2).
As the above results demonstrated, the simulated and observed particulate and total N losses at
the catchment outlet indicated good agreement despite the slight underestimation of the
individual large values when heavy precipitation occurred.



**3.3 Components of the simulated TN and PN at the catchment outlet**

The monthly components of TN and/or PN simulated from the original and upgraded CNMM-DNDC model during the model validation of 2008 at the catchment outlet were illustrated in Fig. 4. Among the TN components including PN, $NH_4^+$, dissolved organic nitrogen (DON) and $NO_3^-$, the simulation from both of the original and upgraded CNMM-DNDC demonstrated that the proportion of $NO_3^-$ at the catchment outlet was larger than that of $NH_4^+$ during the period from May to September when the larger precipitations appeared. Moreover, the upgraded CNMM-DNDC demonstrated that the PN accounted for up to 16.2%−26.6% of the TN components during the period with larger precipitations. Meanwhile, the humads N accounted for 11.3%−20.3% of the PN components, though the humus N accounted for the largest of the PN components. In addition, compared with the original model, the upgraded model simulated the observed TN with smaller nRMSE (42.55% versus 51.67%), better NSI (0.86 versus 0.80) and improved $r^2$ of the ZIRs (0.98 versus 0.90) though no significant difference was found of the ZIRs between the original and upgraded model (Fig. 4).

**3.4 Spatial distributions of sediment yield and particulate C, N, and P losses**

Figure 5 illustrated the simulated spatial distributions of the sediment yield and particulate C, N, and P losses and the effects of different land uses on those in the validation year 2008. The annual accumulated sediment yield simulated by the upgraded model amounted to 0−106.6 t ha$^{-1}$ yr$^{-1}$ with an average of 5.0 t ha$^{-1}$ yr$^{-1}$ in 2008, which was a moderate rainfall year (952 mm) with eight large rainstorm events (exceeding 50 mm rainfall within 24 hours). The simulated annual accumulated particulate C, N, and P losses yielded 0−595.7 kg C ha$^{-1}$ yr$^{-1}$,



0−56.0 kg N ha$^{-1}$ yr$^{-1}$, and 0−7.9 kg P ha$^{-1}$ yr$^{-1}$ with averages of 63.6 kg C ha$^{-1}$ yr$^{-1}$, 6.1 kg N
ha$^{-1}$ yr$^{-1}$ and 0.9 kg P ha$^{-1}$ yr$^{-1}$, respectively. The sloping cultivated cropland areas contributed
to the greatest losses of sediment and particulate C, N, and P nutrients, with 68%, 60%, 58%
and 57% of the total, respectively. Approximately 21% of sediment loss came from the
residential areas as the second largest contributor to sediment loss, while the forest areas were
the secondary sources to particulate C, N, and P losses, with 30%, 32%, and 32% of total losses,
respectively. Meanwhile, the highest rates of the particulate C, N, and P losses per unit area
occurred in the sloping cultivated cropland areas, with 84.1 kg C ha$^{-1}$ yr$^{-1}$, 7.7 kg N ha$^{-1}$ yr$^{-1}$
and 1.1 kg P ha$^{-1}$ yr$^{-1}$, respectively. However, the residential areas yielded to the highest rates
of sediment, i.e., 8.6 t ha$^{-1}$ yr$^{-1}$. The second largest loss rates per unit area of the particulate C,
N, and P appeared in the residential areas. These results demonstrated that sediment yield and
particulate C, N, and P losses caused by surface runoff in the Jieliu catchment were directly
relevant to the type of land use, and the sloping cultivated cropland area became the primary
source of sediment yield and particulate C, N, and P losses. Meanwhile, sediment and
particulate C, N, and P losses from the residential areas could not be neglected.

Moreover, the upgraded CNMM-DNDC model coupled the biogeochemical processes

with soil erosion, which was able to predict the crucial variables relevant to biogeochemical
processes, including the productivity, greenhouse gases, contaminated gases and NO$_3^-$ loss and
the variables related to soil erosion, including the losses of sediment and particulate C, N and P
(Fig. S3, Text S1).





**3.5 Sediment yield and particulate C, N, and P losses under different scenarios**


The simulated results of the single-factor change scenarios of precipitation and
temperature were presented in Figure 6. The sediment yield and particulate C, N, and P losses
(i.e., the target variables) increased with precipitation or air temperature which was reflected by
the positive values. The more positive the slope value is, the greater the target variables
increase and vice versa. The slopes between the air temperature changes of the higher and
lower cooling and warming scenarios and the sediment yield changes yielded −1.25, −1.00,
−0.38 and −0.40, respectively. Compared to the slopes of the lower warming scenarios and the
lower cooling scenarios, the slopes of the higher warming scenarios and the higher cooling
scenarios provided 21% and 5% higher yields of sediment, respectively. Meanwhile, the
changes in particulate C, N, and P losses provided similar but stronger responses to the higher
cooling scenarios. However, the particulate nutrient losses showed a complicated response to
the warming scenarios. For the lower warming scenarios, the particulate nutrient losses
increased with air temperature. The changes in the particulate nutrient losses provided an
increasing tendency in response to the increase in air temperature. In terms of the higher
warming scenarios, the particulate nutrient losses were still increasing, but the rates of increase
rate decreased. These results proved that the increase in air temperature decreased the losses of
sediment but increased the particulate C, N, and P losses, although the promoting effect
became weaker for the higher warming scenarios.
The slopes between the precipitation changes of the higher and lower rain-reduced and
rain-enhanced scenarios and the sediment yield changes resulted in the values of 0.27, 0.37,





0.52 and 0.65, respectively. In comparison with the lower rain-enhanced and rain-reduced
scenarios, the slopes of the higher rain-enhanced and rain-reduced scenarios provided 24%
higher and 34% lower yields of sediment, respectively. Meanwhile, the changes in particulate
nutrient losses provided similar but weaker responses to the changes in precipitation. The
above results demonstrated that the losses of sediment and particulate nutrients increased with
the increasing precipitation. In addition, the contribution from such an elevation role of
precipitation tended to be stronger for the higher rain-enhanced scenarios. Furthermore, the
changes in sediment and particulate C, N, and P losses were more sensitive to precipitation
scenarios than to temperature scenarios.

Table 2 illustrated the results of the multifactor change scenarios and the land use change

single-factor scenario (FL scenario). Compared to the baseline scenario, the FL scenario
reduced stream flow, sediment yield, and particulate nutrient losses by −12.2%, −3.6%, −5.6%,
−7.0%, and −7.2%, respectively. In comparison with the baseline scenario, the low GHG
emissions scenario with air temperature increasing by 1.5 ℃ and precipitation increasing by 10%
increased the stream flow, sediment yield and particulate C, N, and P losses by 21.2%, 4.1%,
5.3%, 5.3% and 5.3%, respectively. The increasing effects of the high GHG emissions
scenarios on the sediment and particulate nutrient losses were more than three times those of
the low GHG emissions scenarios. The low GHG emissions under the FL scenario increased
the stream flow and sediment yield by 5.2% and 0.2%, respectively, but decreased the
particulate C, N, and P losses by −0.8%, −2.3%, and −2.5%, respectively. Moreover, the high
GHG emissions under the FL scenario increased the stream flow, sediment yield, and





particulate C, N, and P losses by 47.9%, 9.2%, 9.3%, 7.8%, and 7.7%, respectively. The
short-term and long-term no-tillage scenarios decreased the losses of particulate nutrients by
approximately 1% and 20%, respectively, but provided no effect on sediment yield compared
with the baseline scenario (data not shown).
**3.6 Relationship among the variables relevant to soil erosion, productivity and C/N losses**

Figure 7 illustrated the relationships between the variables relevant to soil erosion and

biogeochemistry for different land use types. No soil erosion in the RF crop system because of
the year-round flooding regime. For the other three land use types, the significant positive
correlations ($r > 0.88$) between sediment yield and particulate nutrients were found, because
they were entrained by water and moved with water flow. With regard to the SU crop system,
the particulate nutrients were significantly correlated with $NO_3^-$ losses through leaching ($r >$
0.6), though the correlation coefficient between sediment yields and $NO_3^-$ losses through
leaching only yielded to 0.26 (insignificantly). For the SP crop system, the variables related to
soil erosion (including sediment yields and particulate nutrients) were negatively correlated
with $NH_3$ emissions ($r > 0.65$), while they were positively correlated with $NO_3^-$ losses through
runoff ($r < -0.61$). As to FL, significantly positive correlations between the variables related to
soil erosion and $NO_3^-$ losses through leaching/runoff were found ($r > 0.72$).
**4. Discussion**
**4.1 Effect of land use on soil erosion and particulate C, N, and P losses**

Land use change has been considered one of the most important factors affecting the

intensity and distribution of surface runoff and soil erosion (Dunjó et al., 2004; Kosmas et al.,
1997; Wei et al., 2007; Zhang et al., 2021b). Our study also provided consistent results, which



indicated that the intensity of soil erosion and the corresponding particulate C, N, and P losses
in the Jieliu catchment were closely related to land use, with the following order: sloping
cultivated cropland > residential area > forest land. There were three major reasons why forest
land contributed to the lowest losses of sediment and particulate nutrients among the above
three land uses. First, canopy interception reduced the amount of rainfall reaching the ground,
which directly decreased the occurrence of runoff and subsequent erosion (Greene and Hairsine,
2004; Hou et al., 2020; Vasquez-Mendez et al., 2010). Several previous studies also reported
that forest land with a thick canopy exhibited a lower amount of runoff than did other land uses
(Mehri et al., 2018; Mohammad and Adam, 2010; Nunes et al., 2011). Fortunately, the direct
protection mechanism by canopy interception was involved in the CNMM-DNDC model,
which was calculated using the leaf area index (Zhang et al., 2018). Second, the litter cover of
forest land protects the soil surface from the direct splash and detachment of raindrops, which
can decrease the formation of mechanical crusts and increase the infiltration capacity and
hence diminish the potential for surface runoff and soil erosion (Casermeiro et al., 2004;
Lemenih et al., 2005; Wainwright et al., 2002). However, the CNMM-DNDC did not take the
protection of litter cover on the soil surface into consideration. Further observation data and
studies are needed to introduce the mechanism of the effect of litter cover on surface runoff and
soil sediment into the CNMM-DNDC model. Last, forest land is equipped with higher soil
organic matter and hydraulic conductivity than other land uses, which can indirectly enhance
soil infiltration and reduce surface runoff (Abrishamkesh et al., 2011; Fu et al., 2000; Lemenih
et al., 2004). The excellent soil properties of forest land soil (e.g., higher soil organic matter



and vertical saturated hydraulic conductivity) have been involved in the CNMM-DNDC model
inputs. Moreover, as the forest litterfall returned to the soil and participated in further C and N
cycling, the content of soil organic matter was enhanced and accumulated. With regard to the
scenario analysis, we found that the scenarios related to FL contributed to greater decreases in
sediment yield than surface runoff (Table 2). The results of the lysimetric plot experiments by
Chen et al. (2012) also demonstrated that vegetation types and human interference had a
relatively small impact on surface runoff but had an appreciable effect on sediment yield.
The canopy of the cultivated cropland served as a weaker hindrance to rainfall than that of
the forest canopy, which suffered from more surface runoff. Furthermore, frequent agricultural
activities (i.e., tillage) loosen the subsurface soil and nutrients, which raises the risk of soil
erosion and the subsequent loss of particulate nutrients (Gregorich et al., 1998; Moldenhauer et
al., 1967; Muukkonen et al., 2009). The CNMM-DNDC model has taken the vertical mixing
effect of tillage on the upper and lower soils into consideration, and this process left the
subsurface soil organic nutrients unprotected and prone to erosion. This explained the
reduction in particulate C, N, and P nutrient losses under the no-tillage scenarios (data not
shown). However, several studies found that tillage disturbed the soil structure and pore size
distribution (Carof et al., 2007; Castellini and Ventrella, 2012; Kay and VandenBygaart, 2002;
Nunes et al., 2010), which made the effect of agricultural activities on surface runoff and soil
erosion difficult to model (Leitinger et al., 2010). Given that the effect of tillage on soil was not
considered in the CNMM-DNDC, the yields of surface runoff and sediment resulting from the
no-tillage scenario were not decreased compared with the baseline scenario with tillage.



**4.2 Effect of climate change on soil erosion and particulate C, N, and P losses**

In past decades, the frequent occurrence of warming and extreme weather events (e.g., extreme precipitation events) has been irrefutable (IPCC, 2019). From 1998 to 2021, the observed annual average air temperature and annual precipitation in the Jieliu catchment also presented an increasing trend but did not have a significant regression relationship (Fig. S4). The reason of why the simulated soil erosion responded to the air temperature changes might be that the climate-sensitive vegetation growth affected the $C_v$ which was the effect factor of the soil erosion in Eq. 1. The asymmetric response of sediment yield and particulate nutrient losses to the cooling and warming scenarios might result from the different response of the vegetation growth to the cooling and warming of air temperature. Our results of the scenario analysis indicated that the losses of sediment slightly decreased with the scenarios treated with climate warming alone, which lay in the higher $C_v$ caused by the enhanced vegetation growth (Ficklin et al., 2009; Zhang et al., 2020a; Zhou et al., 2003). We found that the decreasing effect of increasing air temperature on sediment loss decreased (especially for the scenario with an air temperature increase of 4 ℃, Fig. 6), which might be because the enhanced effect of increasing air temperature on vegetation growth is not unlimited. Once the air temperature exceeds the threshold of the optimum temperature for photosynthesis and vegetation growth, it would have a negative or even harmful impact on plant growth (Chapin, 1983; Schlenker and Roberts, 2009). The complex response of the particulate C, N, and P losses to air temperature increased, probably because they increased with ER and sediment yield, but the ER decreased with sediment. Therefore, the slightly increasing sediment with increasing air temperature and





510 the corresponding decreasing ER might lead to upward or downward fluctuations in particulate

511 C, N, and P losses. However, we found that the rate of soil loss increased with increasing

512 precipitation amount and the corresponding increase in heavy rain events. Jiang et al. (2017)

513 also found that the increase in sediment loss was amplified by the increased precipitation,

514 which was directly accompanied by a dramatic and sustained increase in surface runoff.

515 Therefore, the higher GHG emissions scenarios, in which the soil erosion provided a higher

516 increase response to the rising precipitation and a lower and smaller decrease response to the

517 rising air temperature, might provide a greater risk of soil erosion than the low GHG emissions

518 scenario. Overall, our results indicated that the hydrology of the Jieliu catchment is very

519 sensitive to potential future climate changes, especially to the higher GHG emissions scenarios.

520 **4.3 Interactive effect of climate and land use change on soil and nutrient losses**

521 Changes in either climate or land use imply considerable influences on water and nutrient

522 cycles in a catchment or region (Labat et al., 2004; Milliman et al., 2008; Piao et al., 2007; Yin

523 et al., 2017). Our simulated results indicated that the reduction extent of the scenario with FL

524 on soil erosion, especially on sediment yield and subsequent nutrient losses, offset the

525 increasing extent caused by the low GHG emissions scenario. However, the scenario with FL

526 was insufficient to totally offset the sediment and particulate C, N, and P losses caused by the

527 high GHG emissions scenario. Nevertheless, vegetation restoration might still be able to slow

528 the soaring process of soil erosion caused by climate change in the future. Previous studies

529 primarily focused on the effects of human activity and climate change on the changes in

530 surface runoff or stream flow. Wang et al. (2016) demonstrated that human activity contributed



to slightly larger effects on stream flow changes than climate (59% versus 41%) by analyzing
the long-term records of hydrological data in the Luan River basin in North China. The results
in the Heihe River basin in Northwest China showed that human activities were the dominant
contributor to the variation in runoff in the upper and middle reaches when compared to
climate change (Qiu et al., 2015). However, other studies have shown that the influence of
climate change on soil and water loss was greater than that of human activities. Jiang et al.
(2017) pointed out that climate change, in comparison with anthropogenic activities, was the
primary factor causing the changes in either stream flow or sediment discharge in the Yellow
River basin and Yangtze River basin in China. The Huron River catchment in southeastern
Michigan in the U.S.A. was more sensitive to climate change than to land use change, as
demonstrated by Barlage et al. (2002).

Furthermore, we found that the promoting impacts of both high and low GHG emissions

scenarios on surface runoff were greater than those on sediment yield and subsequent
particulate nutrient losses. In contrast, the reduction effect of the scenarios with FL on sediment
yield and subsequent particulate nutrient losses was stronger than that on surface runoff (Table
2). These results demonstrated that human activity, e.g., the conversion from cropland with
intensive human disturbance to forest land, resulted in a greater mitigation effect on sediment
yield and subsequent particulate nutrient losses than on surface runoff. Therefore, further
studies should consider the effects of human activity and climate change on surface runoff and
on soil erosion as well as the subsequent nutrient losses. In summary, reasonable human
intervention, such as rational land use change, is expected to be a feasible practice to decelerate



soil erosion and subsequent particulate nutrient losses without altering and disturbing the
hydrological cycle of a catchment in the context of global warming.

## Conclusions

The hydro-biogeochemical model (CNMM-DNDC) was improved by introducing the soil

erosion physical model (adopted from the simplified ROSE model) and the element (i.e.,
carbon, nitrogen and phosphorus) enrichment module to estimate soil erosion and the
movements of particulate nutrients. The comparability between the simulation and observation,
including surface runoff, sediment yield, and particulate nitrogen and phosphorus losses at the
lysimetric plot and the stream flow, sediment yield, and particulate N loss at the outlet of Jieliu
catchment, demonstrated that the upgraded CNMM-DNDC model could reliably simulate soil
erosion and the consequential particulate nutrient losses. The spatial distribution characteristics
of sediment yield and the consequential particulate carbon, nitrogen and phosphorus losses
were directly related to the spatial distribution of land use type, among which the sloping
cultivated cropland areas contributed to the greatest losses. The analysis of climate
single-factor change scenarios implied that the high GHG emissions scenarios provided a
greater potential risk of soil erosion, which resulted in the larger soil erosion rates than those in
the low GHG emissions scenarios. The scenarios with all non-forest land changes into forest
land decreased stream flow, sediment yield and particulate C, N, and P losses compared to the
baseline scenario. Anthropogenic activities (e.g., land use change) might be expected to help
mitigate the processes of soil and water losses accelerated by climate change in the future.

## Code and data availability

The CNMM-DNDC model was originally developed by the Institute of Atmospheric



Physics using C++ language, which can be run on a standard PC. The upgraded model is
available on the FigShare (https://doi.org/10.6084/m9.figshare.20210546).

**Author contribution**


Siqi Li arranged data, improved model and implemented the simulation, prepared the
original draft. Yong Li, Xunhua Zheng, Wei Zhang developed the conceptualization and
methodology of this study. Bo Zhu, Pengcheng Hu, Jihui Fan, Tao Wang collected and arranged
data. Shenghui Han, Rui Wang, Kai Wang analyzed data and verified the results. Zhisheng Yao,
Chunyan Liu improved the conceptualization and writing.

**Acknowledgments**


This work was supported by the Chinese Academy of Sciences (grant number:
ZDBS-LY-DQC007; XDA23070100), the special fund of State Environmental Protection Key
Laboratory of Formation and Prevention of Urban Air Pollution Complex
(SEPAir-2022080590), the National Key Scientific and Technological Infrastructure project
"Earth System Science Numerical Simulator Facility" (EarthLab), the National Natural Science
Foundation of China (grant number: 41907280, U22A20562), and the China Postdoctoral
Science Foundation (grant number: 2019M650808).

**Competing interests**


The authors declare that they have no conflict of interest.

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





Table 1 Performance of the revised CNMM-DNDC model in simulating the stream flow,
sediment, and particulate and total nitrogen (N) losses at the Jieliu catchment outlet from 2007 to
2008. Total N refers to the total amount of $NH_4^+$, $NO_3^-$, dissolved organic N and particulate N.

| Variables | Operation | Size | nRMSE | NSI | ZIR | | |
|---|---|---|---|---|---|---|---|
| | | | | | Slope | $R^2$ | $p$ |
| Stream flow | Calibration | 12 | 18.29 | 0.98 | 0.95 | 0.98 | < 0.001 |
| | Validation | 12 | 34.57 | 0.89 | 0.85 | 0.94 | < 0.001 |
| Sediment loss | Calibration | 12 | 34.02 | 0.94 | 0.91 | 0.96 | < 0.001 |
| | Validation | 12 | 38.23 | 0.89 | 0.88 | 0.92 | < 0.05 |
| Particulate N loss | Calibration | 12 | 49.45 | 0.87 | 0.93 | 0.88 | < 0.001 |
| | Validation | 12 | 57.75 | 0.74 | 0.80 | 0.85 | < 0.001 |
| Total N loss | Calibration | 12 | 56.98 | 0.86 | 1.21 | 0.90 | < 0.001 |
| | Validation | 12 | 42.55 | 0.86 | 0.96 | 0.86 | < 0.001 |

The statistical criteria used to quantify the discrepancy between observations and simulations
include the normalized root mean square error (nRMSE), the Nash–Sutcliffe index (NSI) and the
slope, determination coefficient ($R^2$) and significance level ($p$) of the zero-intercept univariate
linear regression (ZIR). Size represents the sample size.






Table 2 Simulated comprehensive effects of precipitation, air temperature and land use change
on surface runoff, sediment yield, and particulate carbon (C), nitrogen (N) and phosphorus (P)
losses in the validation year of 2008. The low greenhouse gas (GHG) emission scenario
represents the scenario of air temperature increasing by 1.5 ℃ and precipitation increasing by
10%. The high GHG emission scenario represents the scenario of an air temperature increase of
4 ℃ and a precipitation increase of 30%. The FL scenario is the abbreviation of the scenario of
upland change into forest land.

| Scenario | Change between the scenario and the baseline (%) | | | | |
| --- | --- | --- | --- | --- | --- |
| | Surface runoff | Sediment yield | Particulate C | Particulate N | Particulate P |
| Low GHG | 21.2 | 4.1 | 5.3 | 5.3 | 5.3 |
| High GHG | 72.9 | 14.8 | 17.8 | 18.0 | 18.1 |
| FL | −12.2 | −3.6 | −5.6 | −7.0 | −7.2 |
| Low GHG with FL | 5.2 | 0.2 | −0.8 | −2.3 | −2.5 |
| High GHG with FL | 47.9 | 9.2 | 9.3 | 7.8 | 7.7 |






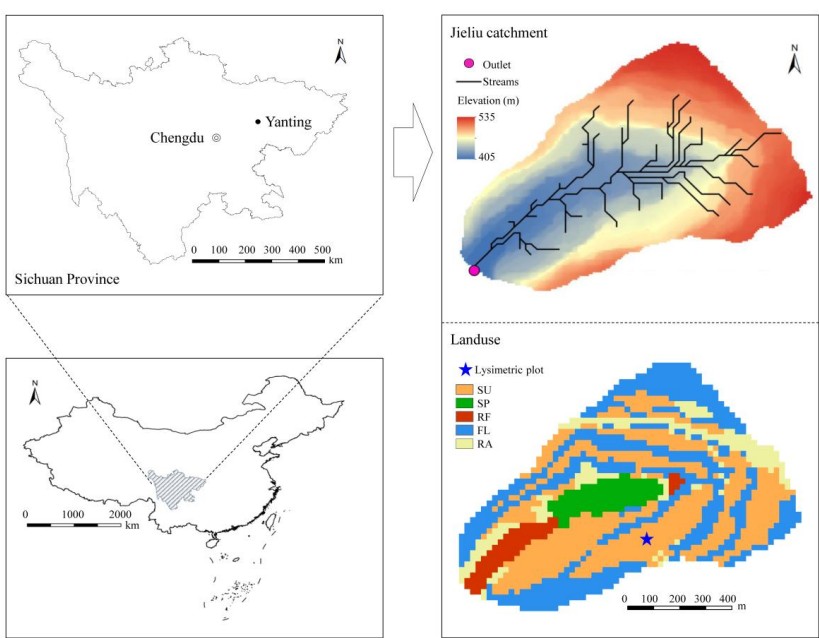


Fig. 1 The location, digital elevation model and land use types of the Jieliu catchment. The land

use types are the sloping uplands with the summer maize−winter wheat rotation (SU), seasonally

waterlogged paddy with the paddy rice−winter wheat rotation or paddy rice−rape rotation (SP),

the winter-flooding paddy with the paddy rice-flooding fallow regime (RF) and the forest land

(FL).

888


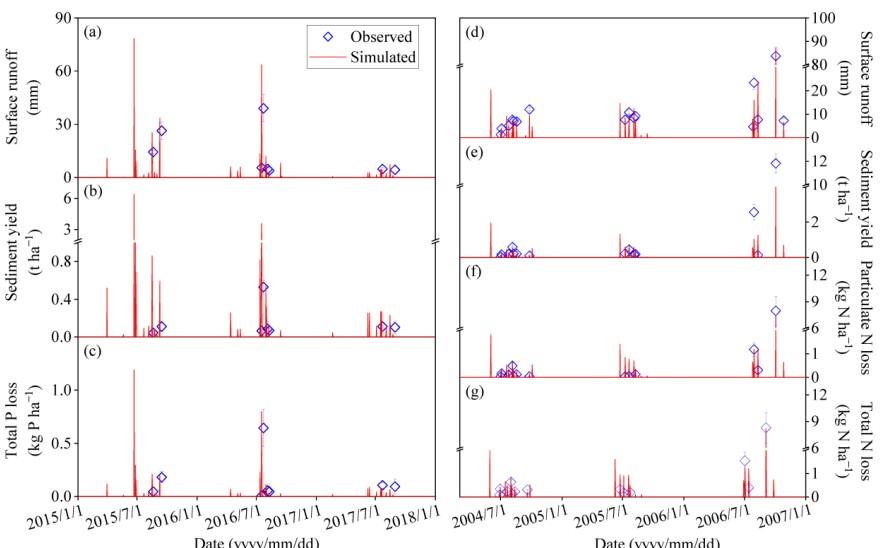

889

Fig. 2 Observed and simulated surface runoff (a), sediment yield (b) and total phosphorus (P)

losses (c) from 2015 to 2017 and surface runoff (d), sediment yield (e), particulate nitrogen (N)

loss (f) and total N loss (g) from 2015 to 2017 in the lysimetric plot. Total P refers to the

dissolved and particulate P. Total N refers to the total amount of $NH_4^+$, $NO_3^-$, dissolved organic

N and particulate N. The vertical bars indicate the standard error of three spatial replicates. The

observed data cited from Deng et al. (2011), Zhang et al. (2018), Li et al. (2022) and Hu (2020)

were provided by Bo Zhu.

897



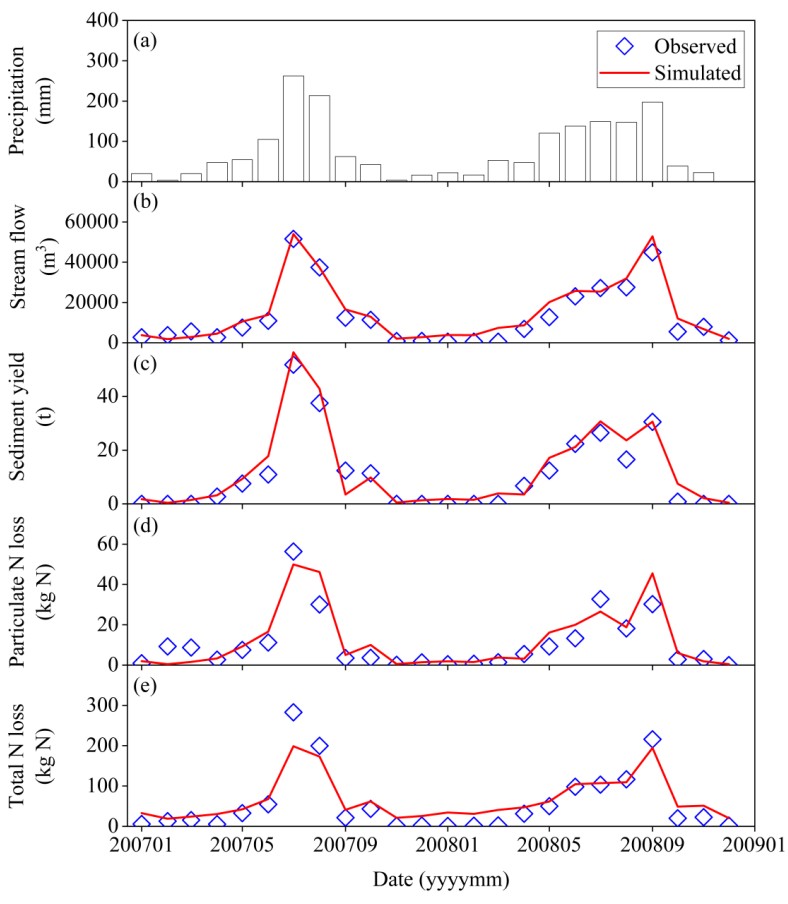

898

Fig. 3 Monthly observed precipitation (a), observed and simulated stream flow (b), sediment

yield (c), particulate nitrogen (N) loss (d) and total N loss (e) at the outlet of the Jieliu catchment

from 2007 to 2008. Total N refers to the total amount of $NH_4^+$, $NO_3^-$, dissolved organic N and

particulate N. The observed data cited from Deng et al. (2011) and Zhang et al. (2018) were

provided by Bo Zhu.

904





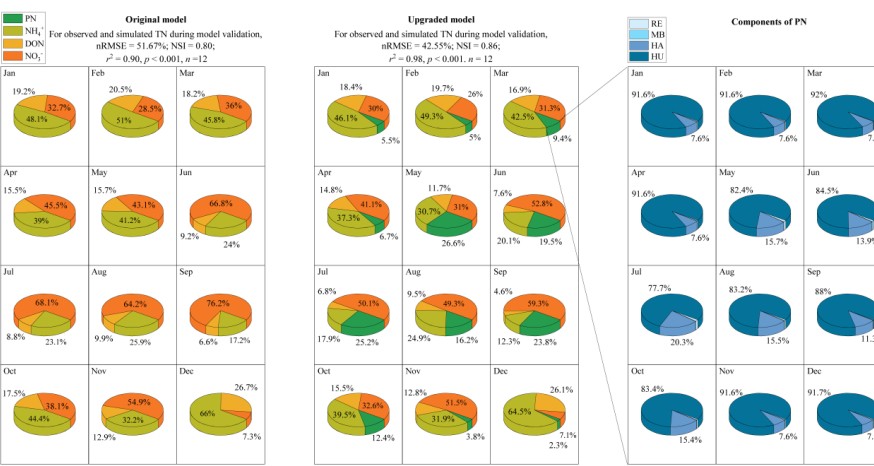

Fig. 4 Components of the simulated total nitrogen (TN) and/or particulate N (PN) of the original and upgraded CNMM-DNDC model during the model validation. DON is the abbreviation of the dissolved organic nitrogen. The components of PN are the N from residue (RE), microbe (MB), humads (HA) and humus (HU).

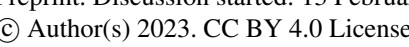

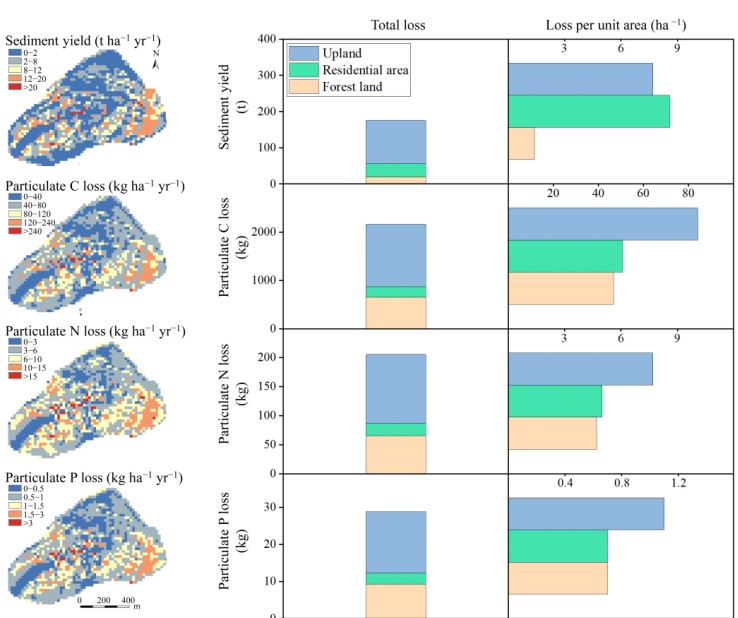

Fig. 5 Simulated spatial distributions of sediment yield, particulate carbon (C), nitrogen (N) and

phosphorus (P) losses and the effects of different land uses (i.e., cropland, residential area and

forest land) in the validation year of 2008.



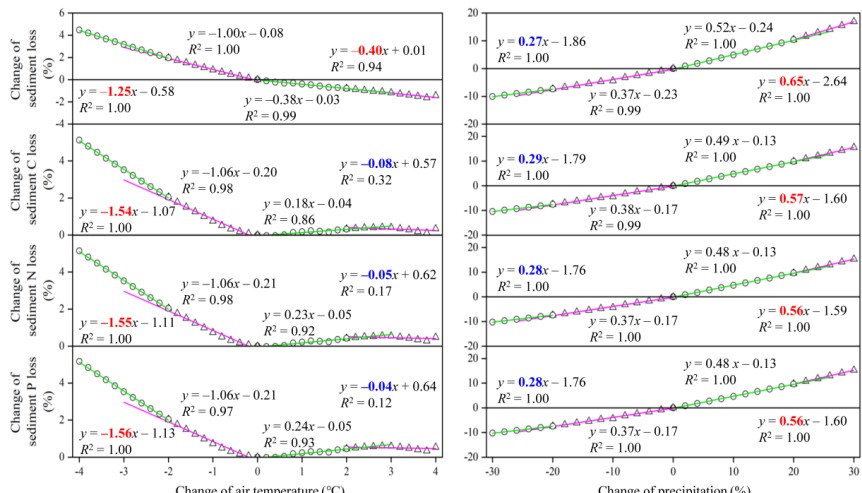

916

Fig. 6 Simulated effects of precipitation and air temperature change on sediment yield and

particulate carbon (C), nitrogen (N) and phosphorus (P) losses in the validation year of 2008.

The air temperature and precipitation single-factor scenarios were divided into four sets. The

scenarios with air temperature reductions and increases 0℃~2℃ and greater than 2℃ were

defined as the lower and higher cooling and warming scenarios, respectively. Similarly, the

scenarios with precipitation reductions and increases 0%~20% and greater than 20% were

defined as the lower and higher rain-reduced and rain-enhanced scenarios, respectively. The

numbers in blue and red in front of the letter $x$ represent that the higher warming or cooling

scenarios (or the higher rain-enhanced or rain-reduced scenarios) result in more and lower

effects on sediment yield and particulate C, N and P losses than the lower ones, respectively.

927

928

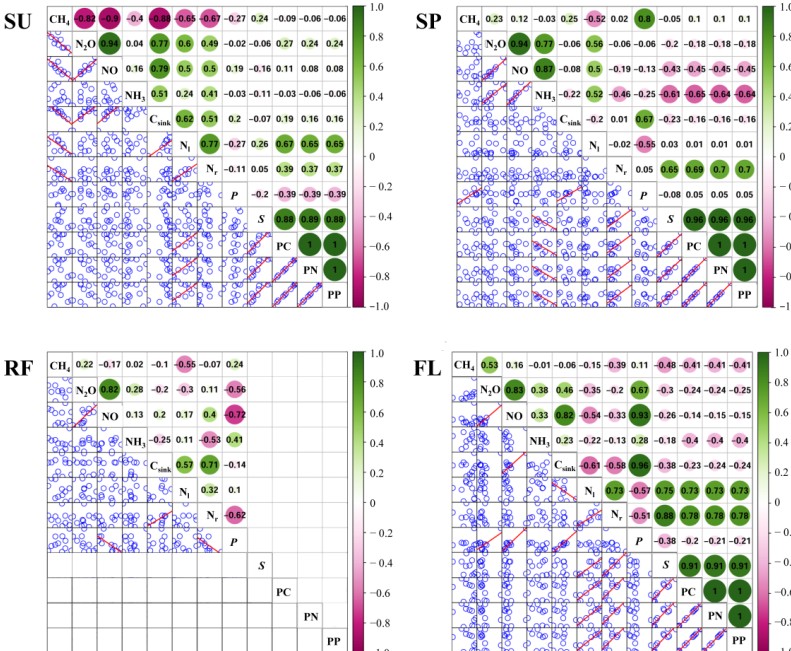

929

Fig. 7 Correlation analysis among the simulated sediment (S), particulate carbon (PC), nitrogen

(PN) and phosphorus (PP) losses, productivity ($P$), C sink density ($C_{sink}$), methane ($CH_4$), nitrous

oxide ($N_2O$), nitric oxide (NO) and ammonia ($NH_3$) emissions, losses of nitrate through leaching

($N_l$) and surface runoff ($N_r$) for different land use types. The land use types are the sloping

uplands with the summer maize−winter wheat rotation (SU), seasonally waterlogged paddy with

the paddy rice−winter wheat rotation or paddy rice−rape rotation (SP), the winter-flooding

paddy with the paddy rice-flooding fallow regime (RF) and the forest land (FL). No losses of $S$,

PC, PN, and PP in the RF crop system because of the year-round flooding regime. The figures in

the circles stand for the correlation coefficients. The correlations with the level of $p < 0.05$ are

considered as significant and the linear regression lines are exhibited.