# Peer review of "Enabling a process-oriented hydro-biogeochemical model to simulate soil"

_Biogeosciences, 2022_

## Author Response (AR1)

**Response to the comments**

**A brief introduction to the revision of the manuscript (MS) as follows:**

The two reviewers' comments were fully incorporated into the revised MS accordingly. The text of the MS was revised by: i) reorganizing and rewriting the Materials and methods section by providing more details about model modifications and observation methods and by adding Table S3 to sum up the scenarios and shorten the long description subsection about scenario setting; ii) adding the consideration of the productivity in the scenario analysis and the corresponding results and discussion; iii) further discuss the effects of the air temperature change on the vegetation growth and soil erosion.

Furthermore, we recompiled the Supplementary materials by adding Table S1 to present the soil properties of different soil profiles for different land use types, Table S2 to list the information on the observation data applied for model calibration and validation and Table S4 to list the examples for the effects of temperature on biogeochemical processes in CNMM-DNDC model. Moreover, the schematic of the existing and upgraded modules (i.e., Fig. S1), the results of the simulated effects of no-tillage on soil erosion (i.e., Fig. S5), the simulated annual surface runoff of different land use types (i.e., Fig. S6), and the effects of the air temperature change on the vegetation growth and soil erosion (i.e., Fig. S8) were added to support our results and discussion.

*Reply on associate editor*

*Dear authors:*

*Thank you for making the meaningful open discussion.*

*The two reviewers found good scientific significance in your manuscript, although there remains room for improvement in scientific and presentation qualities. Both referees required more clarification in methodological descriptions about observational data and model calibration/evaluation. Therefore, I conclude that the manuscript needs major revisions on the basis of your reply comments before making a final decision.*

>> Agreed and revised. The two reviewers' comments were fully incorporated into the revised MS accordingly.

i)    For the observation data, Table S2 was added to to list the information on the observation data applied for model calibration and validation in the revised Supplementary. And the detailed description about the method of the lysimeter plot was added in the subsection 2.1. Please see changes in Lines 142−147 in the revised MS.

ii)   For model calibration/evaluation, we reorganized and rewrote the section of 2.3 Model modifications to make clear about the model calibration description. Please see changes in Lines 159−231 in the revised MS.

*General comments*

*The improved CNMM-DNDC model enables prediction of water and soil particles and the amount of C, N, and P losses that move with them. The temperature, precipitation, and land use scenario analyses conducted using this model may provide useful information for prioritizing measures for vulnerable areas that are susceptible to soil erosion. The focus of the paper is clear and well organized. However, because the observations in this study were cited from several papers, some of the descriptions were confusing to the reader because they did not fully explain the observation methods or were indistinguishable from the results of this study. Some additions and corrections are necessary and are listed below. I hope you will find them helpful.*

**>> Agreed and revised. The observation data, which were applied for model calibration and validation, were summarized in the Table S2 in the revised Supplementary. And the description about the method of the lysimeter plot was added in the subsection 2.1. Please see changes in Lines 142−147 in the revised MS. Furthermore, the original Figure S1 and Table S1 and the corresponding results in sebsection 3.1 derived from Li et al. (2022) were removed. Please see changes in Lines 388−392 in the revised MS and the revised Supplementary.**

*Abstract*

*L32-33 Another important result is that the upgraded model now estimates more particulate N in TN during periods of high precipitation compared to the original.*

**>> Response and revised. The original CNMM-DNDC had no capacity to predict the particulate N loss, so we rewrote this comment as "The upgraded CNMM-DNDC demonstrated that more proportion of the particulate N to total N during the period with large precipitations than that during the droughty period (16.2%−26.6% versus 2.3%−12.4%)" to incorporated in the revised MS. Please see changes in Lines 33−37 in the revised MS.**

*L37 It would be necessary to explain that land use change here means returning the land from an anthropogenic environment to natural vegetation.*

**>> Agreed and revised. Please see changes in Line 41 in the revised MS.**

*1. Introduction*

*L91 There is no description of a biogeochemical module for phosphorus; does this mean using CNMM's module for phosphorus?*

**>> Revised. Please see changes in Line 103−104 in the revised MS.**

L97 Throughout the paper, it seems inadequate the description of productivity is inadequate (especially, L429~). How would you determine the accuracy of the productivity prediction? Productivity seems to have no or negative impacts on S, PC, PN, and PP (Fig. 7), but the trade-off between reducing environmental impact and maintaining or increasing productivity would be better considered in the scenario

analysis.

**>> Agreed and revised.**

**We added some the description about the relationship between the productivity and the variables relevant to soil erosion. Please see changes in Lines 542−545 in the revised MS.**

**The accuracy of the productivity and other related variables prediction in this catchement had been evaluated by Zhang et al. (2018), which was addded. Please see changes in Lines 293−297 in the revised MS.**

**Meanwhile, the productivity was added to be considered in the scenario analysis and some results and discussion about productivity were added. Please see changes in Lines 353−354, 488−491, 504−508 and 517−519, Table 2 and Fig. 6 in the revised MS.**

*L 106-107 There is a description of a soil erosion physics model, but no description of an elemental enrichment module in Introduction section.*
**>> Revised. Please see changes in Line 90−92 in the revised MS.**

**2. Materials and methods**
*L126 For the parameter $S_l$ in equation 1, it would be better to include a slope angle. Also, does this mean that the artificial lands accounts for 15% of the 35 ha?*
**>> Revised.**

**The parameter $S_l$ in Eq. 1 was explained by detail. Please see changes in Lines 176−178 in the revised MS.**

**The proportions of the land uses were rechecked and revised in Line 137 in the revised MS.**

*L 137 It would be better to cite Text S1 here.*
**>> Revised. Please see changes in Line 165 in the revised MS.**

*L 172-173 Do you mean each mass balance of water, C, N, and P? Please clarify.*
**>> Revised. Only the mass balance of soil body was ignored. The upgraded model considered the mass balances of soil water and the elements of C, N, and P, without considering soil body balance. Please see changes in Lines 260−261 in the revised MS.**

*L183 Even when citing Deng et al. (2011) or Hu (2020), please add description of observation method by lysimeter plot.*
**>> Revised. The description about the observation methods by the lysimeter plot was added. Please see changes in Lines 142−147 in the revised MS.**

*L191-193 Is there carbon (or nitrogen or phosphorus) flow between the three pools of litter, between the two pools of microbes, and between the?*
**>> Response and revised. The flows of carbon, nitrogen and phosphorus among the pools of the labile and resistant organic and inorganic were considered in the**

**CNMM-DNDC. For example, the carbon and nitrogen of the litter, humads and humus pools and the phosphorus of the pools of the active or passive organic P and the inert stable P were flowed to inorganic pools and the microbe pools by decomposition. Please see changes in Lines 253−257 in the revised MS.**

*L214-216 In relation to Figure 7 and S3 and text S1, please provide a brief description of how the observed data is measured.*
**>> Revised. The description about the observation methods by the lysimeter plot was added. Please see changes in Lines 142−147 in the revised MS.**

*L202 There is a lack of information on soil properties and model parameters by land use. Even if derived from Zhang et al. (2018), It would be useful for the readers to include a table where the soil properties (e.g. soil bulk density, soil depth, soil organic C, N, and P, soil hydraulic conductivity, and Q10 value, ...) are listed as Table S2 in the supplement.*
**>> Revised. The information about soil properties (e.g., soil bulk density, pH, clay content, field capacity, wilting point, saturated hydraulic conductivity, organic carbon, and total nitrogen and phosphorus contents) of different soil profiles for different land use types, which were the input data for driving the model operation, were added and listed in Table S1. Please see changes in Lines 275−277 in the revised MS and Table S1 in the revised supplementary.**

**Response and revision to Q10 value. The effects of temperature on biogeochemical processes were only varied on the temperature itself instead of being set by land use types. We provided some examples for the effects of temperature on the biogeochemical processes. Please see the Table S4 in the revised Supplementary and Lines 609−610 in the revised MS.**

*L230-233 Is this the baseline scenario? This needs to be clarified.*
**>> Revised. Please see changes in Lines 300−304 in the revised MS.**

*L237-238, 240-241 Do you use the abbreviations $T_{air}+$, $T_{air}−$, P+, P− anywhere but here?*
**>> Revised. The abbreviations $T_{air}+$, $T_{air}−$, P+, P− used only once here were removed. Please see changes in Lines 309−310 and 311−312 in the revised MS.**

*L241-242 Is the transition period from cropland to forest considered?*
**>> Response. The scenario analysis was conducted in one year of 2008, so the transition period from cropland to forest was not considered.**

*L255 I assume the Q10 value will have a significant impact on the temperature scenario, but how will it be set by land use (related to your comment on L202)?*
**>> Response and revised. The effects of temperature on biogeochemical processes were only varied on the temperature changes instead of being set by land use types in the CNMM-DNDC. We provided some examples for the effects**

of temperature on biogeochemical processes. Please see the Table S4 in the revised Supplementary and Lines 609−610 in the revised MS.

*L280 closer to 100, not 1?*
>> Corrected. Please see changes in Line 362 in the revised MS.

*3. Results*
*L308-313 Figure S1 shows only total and particulate N loss results. Please add the description of the $NH_4^+$ and $NO_3^-$ results. Are the results in Figure S1 and Table S1 simulated by the model upgraded in this study? The results in Figure S1 are derived from Li et al. (2022), but are described in the result section as if they were from this study, confusing the reader.*
>> Revised. We added the Fig. S2 to exhibit the observed and simulated $NH_4^+$ and $NO_3^-$ losses in the lysimetric plot. And descriptions in the Results, the original Fig. S1 and Table S1 derived from Li et al. (2022) were removed. Please see changes in Lines 388−392 in the revised MS and the revised Supplementary.

*L336-337 Regarding the prediction of total N loss by CNMM-DNDC, does it mean that the model underestimated the loss of $NH_4^+$ (as in L312-313) or dissolved organic N because it successfully predicted the loss of particulate N and $NO_3^-$?*
>> Revised. Please see changes in Lines 423−425 in the revised MS.

*L392-394 Why does the cooling scenario have a larger change in sediment yield and particulate C, N, and P loss?*
>> Response and revised. The explanation on the larger effects of the cooling scenario on sediment yield and the particulate C, N, and P loss was added in Lines 621−622 and 628−630 in the revised MS.

*L429 Losses due to $NO_3^-$ leaching as well as NO3- runoff are correlated with S, PC, PN, and PP in SU and FL land use (Figure 7). There must be a time lag between $NO_3^-$ leaching from the soil layer and the other observations, but how can there be a correlation? An explanation of the observation method of the data would be helpful to the reader.*
>> Revised. The variables relevant to soil erosion, productivity and C/N losses for the relationship analysis were derived from the model simulation. Please see changes in Lines 528−529 in the revised MS. The relationship between the variables related to soil erosion and $NO_3^-$ losses through leaching/runoff might be because all these variables were related to the precipitation. Please see changes in Lines 541−542 in the revised MS. The description about the observation methods by the lysimeter plot was added. Please see changes in Lines 142−147 in the revised MS.

*L435-L441 The values of r are probably incorrect.*
>> Revised. We rechecked the all values of the Pearson correlation coefficient (*r*)

**and added some explanation about the *r*. Please see changes in Lines 369−373 in the revised MS.**

*4. Discussion*

*L479-488 In the CNMM-DNDC model, it is not clear what can and cannot be reproduced by tillage management. Does the current situation mean that there is a vertical mixing effect with respect to soil chemical properties and no such effect with respect to soil physical properties? Results and discussion also cover the "cultivation scenarios" (L425-428, L481-488, etc.), so it would be better to present the results related to the tillage scenario as figures and tables in the supplement.*

**>> Agreed and revised. We rewrote the ambiguous sentences and added the results related to the tillage scenario as Fig. S5. Please see changes in Lines 599−603 in the revised MS and Fig. S5 in the revised Supplementary.**

*L498-501 For example, does vegetation coverage fraction affect evapotranspiration rates, thereby changing soil moisture content and affecting sediment runoff rates? A more detailed explanation would be useful to the reader.*

**>> Agreed and revised. Please see changes in Lines 622−625 in the revised MS.**

*L508 ER is not explained in the text. enrichment ratio?*

**>> Revised. We Please see changes in Lines 640−642 in the revised MS.**

*Figures*

*What is RA, the land use legend in Figure 1? Is it a residential area?*

**>> Revised. Please see changes in the footnotes of Fig. 1 in the revised MS.**

*Figures 4, 5, 6, 7, and S3 are too small to see the text.*

**>> Revised. Figures 4, 5, 6, 7, and S3 were enlarged in the revised MS.**

*Figure 5, upland, or cropland?*

**>> Corrected. Please see changes in Fig. 5 in the revised MS.**

*For the legend in Figure 7, it would be useful to include a description of the circle size and color intensity.*

**>> Revised. Please see changes in Fig. 7 in the revised MS.**

*Tables*

*In Table S1, it is RMSE, not nRMSE.*

**>> Corrected. In the original Table S1, it is nRMSE instead of RMSE. But the original Table S1, cited from Li et al. (2022), was removed.**

*Text S1*

*Fig.7 a-h are not found. Is it Fig. S3?*

**>> Corrected. Please see changes in Text S1 in the revised MS.**

*Reply on Referee #2*

*Synopsis*
*The authors extend the functionality of the existing hydro-biogeochemical land use model CNMM-DNDC with new routines to simulate soil erosion and associated transport of C, N and P in various fractions, which presents a novel enhancement of an existing agro-ecosystem model. The enhanced model is tested for a lysimeter plot and a small catchment outlet showing largely good agreement with measurements. A subsequent scenario analysis reveals sensitivies to changes in precipitation, temperature, and land use whereas expectedly the highest sensitivity is found for the first.*
**>> Thanks.**

*General assessment*
*In principle, the study presents an interesting model enhancement and its evaluation for a case study, combining soil organic matter and nutrient cycling, soil greenhouse gas emissions, and water erosion. However, the manuscript has several shortcomings that render it challenging to assess. First of all, the authors apparently calibrated the model for the study region, but this process is not well recorded in the methods except for few scattered statements, which renders it hard to follow. Also the choice of the erosion model itself is not fully clear as detailed below. Furthermore, it is not clear what capabilities the original and enhanced models have. The authors may consider adding a schematic of key modules included in the existing and enhanced model to provide readers with an overview. Last but not least, the model evaluation methods need to be reassessed and/or very well justified as zero-intercept regressions can be quite misleading. I hence recommend the authors thoroughly revise their manuscript focusing on clear and consistent descriptions of all methods and results.*
**>> Fully agreed and revised the MS accordingly.**
- **i)** **we reorganized and rewrote the section of 2.3 Model modifications to make clear the description about the model calibration . Please see changes in Lines 159−231 in the revised MS.**
- **ii)** **we added more descriptions and Text S2 to provide the reasons why we chose this erosion model. Please see changes in Lines 178−185 in the revised MS and the Text S2 in the revised Supplementary.**
- **iii)** **the new Fig. S1 was added to present the schematic including the existing and upgraded modules. Please see changes in Fig. S1 in the revised Supplementary.**
- **iv)** **the regular linear regressions were applied to conduct the model evaluation instead of the original zero-intercept linear regressions. Please see changes in subsection 2.6 and 3.2 and Table 1 in the revised MS.**

*Specific comments*
*L37 and other places: The authors use land use change as a scenario to mitigate*

*erosion. What the authors do not consider is the study of field management practices to control erosion. This is the far more common approach to control erosion and implemented in various models such as USLE, RUSLE and their derivatives (see e.g. https://doi.org/10.1016/j.envsci.2015.03.012). The authors need to clearly state why still this erosion model was selected and how they expect management options would affect their results.*

**>> Revised.**

**We added some results and discussion related to the tillage scenario. Please see changes in Lines 592−596 and 599−603 in the revised MS and Fig. S5 in the revised Supplementary.**

**More descriptions and Text S2 were provided to give the reasons why we chose this erosion model. Please see changes in Lines 178−185 in the revised MS and the Text S2 in the revised Supplementary.**

*L78: To my knowledge, the SWAT model includes USLE and MUSLE not RUSLE. Check again what the cited model can do and consider better describing the state-of-the-art.*

**>> Revised. Please see changes in Lines 72−73 and 85 in the revised MS.**

*L143-145: First, is there a reference for this statement? Second, the authors earlier justify their choice of the ROSE model based on the inclusion of the three processes (detachment, transport, sedimentation (L72ff)) in equilibrium but state here that they omit two of them. So why not take another model that as well neglects some processes but may have other advantages such as erosion control management?*

**>> Revised. We added the reference for the statement what you mentioned. Please see changes in Line 173 in the revised MS. And more descriptions about the reason why we chose this erosion model were added. Please see changes in Lines 178−185 in the revised MS and the Text S2 in the revised Supplementary.**

*L158: Unclear what the growing index is. Is this similar to LAI? Why not take LAI directly as is done for natural vegetation? Also, does the growing index vary among crops? Differences in soil cover can be substantial between a wide row crop such as maize and dense grass-like crops such as wheat. Commonly, erosion models use crop type-specific coefficients that reflect LAI and plant density (e.g. http://dx.doi.org/10.1016/j.landusepol.2015.05.021) or aboveground biomass (e.g. https://doi.org/10.1098/rstb.1990.0184).*

**>> Revised. We added some explanation about the $C_v$ values of different land use types. Please see changes in Lines 222−231 in the revised MS. However, the different effects on soil erosion and rainfall interception by various crop planting density, e.g., the wide row maize and the dense grass-like wheat needed more observations to modify and evaluate the CNMM-DNDC in future. This part and the two literature mentioned above were added into the revised MS. Please see changes in Lines 586−589 in the revised MS.**

*L183: How was the model calibrated? If this was done systematically, e.g. using auto-calibration, the parameter ranges, method, etc. need to be stated. Also, were single parameters calibrated one by one or in combination?*

**>> Revised. We reorganized and rewrote the section of 2.3 Model modifications to make clear the description about the model calibration. Please see changes in Lines 159−231 in the revised MS.**

*L205: What field management practices are considered and how are they parameterized? Even if these and various other data are the same as in Zhang et al. (2018), they need to be stated here to facilitate understanding of the study. An overview of the models' functionalities would be helpful in this regard.*

**>> Revised. The inputs of the field management practices for the CNMM-DNDC and the schematic of the existing and upgraded modules in the CNMM-DNDC were added. Please see changes in Lines 178−185 in the revised MS and Fig. S1 in the revised Supplementary. Meanwhile, an overview of the model was added as Subsection '2.2 Overview of the CNMM-DNDC model' according to your comments. Please see changes in Lines 148−158 in the revised MS.**

*L228: The scenario section would profit from a table synthesizing the scenario definitions. This may also allow for shortening this quite lengthy section.*

**>> Revised. The scenarios were summed up in Table S3. And the long subsebction of "2.5 Climate and land use scenario settings" was shortened. Please see changes in subsection 2.5 in the revised MS and Table S3 in the revised Supplamentary.**

*L267: Tillage comes here at some surprise as it is not mentioned earlier to be included in the model. Later, the results are only briefly mentioned. Again, it would be good to have an overview of the scenarios and how they are implemented.*

**>> We added some results and discussion related to the tillage scenario. Please see changes in Lines 592−596 and 599−603 in the revised MS and Fig. S5 in the revised Supplementary. And the scenarios were summed up in Table S3. Please see changes in Table S3 in the revised Supplamentary.**

*L277: Why was a zero-intercept regression used and not a regular one? It's for most cases strongly discouraged to force a regression through zero, which may also substantially affect the calculation of the correlation coefficient.*

**>> Revised. The regular linear regressions were applied to conduct the model evaluation instead of the original zero-intercept linear regressions. Please see changes in subsection 2.5 and 3.2 and Table 1 in the revised MS.**

*L292ff: This seems to be part of the calibration methods and should accordingly go to the methods section, ideally a new sub-section focusing on calibration.*

**>> Revised. We reorganized and rewrote the section of 2.3 Model modifications to make clear the description about the model calibration. Please see changes in**

**Lines 159−231 in the revised MS.**

*L367: It's surprising that residential areas are a major source of soil erosion. The authors should explain and show more thoroughly why this is the case here.*
**>> Revised. We added some explanation about why the residential areas acting as a major source of soil erosion. Please see changes in Lines 553−556 in the revised MS and the Fig. S6 in the revised Supplementary.**

*L486: Earlier you mention tillage scenarios; here you state that tillage was not considered. So which of the two is correct?*
**>>Corrected. Please see changes in Lines 599−603 in the revised MS.**

*Table 1: Columns Operation and Size require explanation.*
**>> Revised. Please see changes in the footnotes of Table 1 in the revised MS.**

*Figure 4: It's unclear why the third panel (PN) is connected with lines to March in the central panel. If this is just about showing that the third panel refers to PN, it would be better to remove the lines and simply mention this in the caption.*
**>> Revised. Please see changes in Fig. 4 in the revised MS.**

*L909: I'm not familiar with the term "humad". Do you mean litter? Please make sure to only use terms that are common in the related literature.*
**>> Response and revised. We usually considered the word "humad" as the labile or resistant humus. To improve readability of the MS, the term "humad" was changed to the common one "labile or resistant humus" according to the reviewer's suggestion. Please see changes in Lines 250−251, 438 and 1059−1060 in the revised MS.**

*Figure 5: Top right x-axis unit [ha$^{-1}$] is confusing at first sight. Consider adding the numerator unit as well.*
**>> Revised. Please see changes in Fig. 5 in the revised MS.**

*Figure 6: Unclear what the line colors (green and violet) refer to. Seems like they overlap while the regressions should be mutually exclusive?*
**>> Revised. We added the explanation about the green and violet line. Please see changes in the footnote of Fig. 6 in the revised MS.**

*Figure 7: The authors explain what the circles in the top right half mean (L937) but not what is shown in the corresponding scatter plots of the bottom left. Do these show the data points relating to the regression coefficients? Please include in the caption.*
**>> Revised. Please see changes in the footnote of Fig. 7 in the revised MS.**

*Minor comments:*
*- Language is in principle good but there is a substantial number of terms that*

*somewhat miss the point. The authors should thoroughly check this while I can only provide examples here:*

**>> Response and revised. We also double-checked the whole MS thoroughly and adjusted the inconsistent expressions and terms throughout the revised MS.**

*L21 "subsequent" should be "associated"*

**>> Revised. We changed the mentioned word "subsequent" to "associated" throughout the MS. Please see changes in Lines 21, 51, 95, 114, 116, 123, 163, 282, 284, 560, 591, 657, 676, 678, 681, 683 and 685 in the revised MS.**

*L26: The model name needs to be spelled out*

**>> Revised. Please see changes in Line 26−27 in the revised MS.**

*L31 and various other places: "credible" is not a scientific term.*

**>> Revised. The unscientific term "credible" was changed to the scientific term "acceptable". Please see changes in Lines 33 and 404 in the revised MS.**

*L33: "larger" should be "higher"*

**>> Revised. Please see changes in Line 33 in the revised MS.**

*L40: "may become a" should be "renders it a potential"*

**>> Revised. Please see changes Line 45 in the revised MS.**

*L50: "deteriorate" does not fit here*

**>> Revised. We rewrote the mentioned sentence. Please see changes in Line 55 in the revised MS.**

*L62: "water-reduced" should be "water-induced"?*

**>> Revised. Please see changes in Line 67 in the revised MS.**

*L91 and various other places: "complicated" should better be something like "complex"*

**>> Revised. Please see changes in Lines 93, 99, 106 and 118 in the revised MS.**

*L191: "reliable" should be "labile"?*

**>> Revised. Please see changes in Lines 249−251 in the revised MS.**

*L192: "humads" is not in the dictionary. Is this "litter"?*

**>> Response and revised. We usually considered the word "humad" as the labile or resistant humus. To improve readability of the MS, the term "humad" was changed to the common one "labile or resistant humus" according to the reviewer's suggestion. Please see changes in Lines 250−251, 438 and 1059−1060 in the revised MS.**